# Moving beyond MARCO

**Nicholas Rosa[1,2], Christopher J. Watkins[3¤a], Janet Newman[2¤b]***

**1** Department of Electrical and Computer Systems Engineering, Monash University, Clayton, VIC, Australia, **2** Biomedical Manufacturing, CSIRO, Parkville, VIC, Australia, **3** Scientific Computing, CSIRO, Clayton, VIC, Australia

¤a Current address: Uber Technologies Inc, San Francisco, CA, United States of America
¤b Current address: School of Biotechnology and Biomolecular Sciences, University of NSW, Kensington, NSW, Australia
* janet.newman@unsw.edu.au

**Data Availability Statement:** The data and code are found in the following repositories: C3 Testset: 10.5281/zenodo.4635300 Code: 10.5281/zenodo.7647927 https://paperswithcode.com/sota/image-classification-on-imagenet. September 29, 2022 6/33.

## Abstract

The use of imaging systems in protein crystallisation means that the experimental setups no longer require manual inspection to determine the outcome of the trials. However, it leads to the problem of how best to find images which contain useful information about the crystallisation experiments. The adoption of a deeplearning approach in 2018 enabled a four-class machine classification system of the images to exceed human accuracy for the first time. Underpinning this was the creation of a labelled training set which came from a consortium of several different laboratories. The MARCO classification model does not have the same accuracy on local data as it does on images from the original test set; this can be somewhat mitigated by retraining the ML model and including local images. We have characterized the image data used in the original MARCO model, and performed extensive experiments to identify training settings most likely to enhance the local performance of a MARCO-dataset based ML classification model.

## 1. Introduction

Structural biology, where knowledge of the atomic detail of biological macromolecules informs our understanding of their function, has profoundly changed our understanding of the living world [1]. The most successful technique used to generate the 3D atomic models is X-ray crystallography, where the molecule of interest is coaxed into a well-ordered lattice (a crystal) which is used as a diffraction grating for a monochromatic beam of hard X-rays [2]. The resulting diffraction pattern can be used to generate a model of the molecule of interest. The value of this technique can be seen not only in the number of Nobel Prizes associated with the technique [3], but also in the contribution that X-ray crystallography makes to human health [4, 5].

Every technique has a bottleneck; in macromolecular crystallography the limiting factor is the production of well-ordered crystals which are suitable for diffraction experiments [6, 7]. Diffraction quality crystals need to be sufficiently large (10–100 ¼m per side), single and well-ordered. The process of generating crystals has remained essentially unchanged for many decades—most often a small droplet of a pure and concentrated sample of the macromolecule

**Funding:** The author(s) received no specific funding for this work.

**Competing interests:** The authors have declared that no competing interests exist.

is mixed with a similar volume of a chemical cocktail, and the system is left to equilibrate against a larger volume of the cocktail [8, 9]. The outcome of the experiment is monitored either by observing the droplet directly under a microscope or by looking at an image of the droplet. Crystallisation requires firstly that that the solution being crystallized is or becomes supersaturated, secondly that nucleation (the creation of a transient assembly of molecules into a formation suitable for other molecules to join) occurs and thirdly that the nuclei continue to grow, rather than to dissolve [9]. All three of these processes can be slow and stochastic, thus each droplet/trial has to be monitored repeatedly over time (days to months) for crystal formation or other, less positive outcomes.

Although the fundamental approach to crystallisation has been unchanged for 50 years, the details of how the trials are set up have evolved, largely in response to the Structural Genomics initiatives of the early 2000s [10]. These high-throughput approaches have been adopted by many modern laboratories; particularly larger industrial and service laboratories. Firstly, the volumes of sample and cocktail have shrunk from many microlitres (μL) per trial to 50–1000 nanolitres (nL) per trial, as the crystal size required for diffraction experiments has decreased. This has allowed for trials to be set up in parallel using laboratory automation—often 96–1536 droplets on a single SBS footprint plate (Standards ANSI/SLAS 1–2004 through ANSI/SLAS 4–2004) [11]. In turn, the high density has driven the development of automated incubation/ imaging systems, where crystallisation plates with many trials are stored and periodically imaged—there are generally between 4 and 20 plate inspections (*i.e.* the process by which discrete set of images, one for each trial on a plate, is collected at a single timepoint) over the course of weeks to months. This removes the necessity for multiple manual inspections of the droplet-containing plate, but leads to the generation of large numbers of images that need to be scanned for interesting results [12]. Many of the images in a droplet timecourse will be essentially identical, as much of the droplet variation happens early in the lifetime of the crystallisation experiment [13]. There will still be slight differences between the timecourse images even if the droplet remains unchanged; these are due to subtle variations in lighting or positioning of the droplet under the imaging lens. Early attempts to find interesting results by looking for changes between images in a timecourse were too slow to be useful [14].

The problem of interpreting images was anticipated even before the first commercial imaging systems were available and there were a number of academic groups working on Structural Genomics projects who developed image interpretation applications based on Random Forest or other machine learning (ML) algorithms [15–18]. However, none of these were accurate enough to be adopted outside their home laboratory. More recently, an approach based on the Inception V3 deep learning architecture [19] showed that machine interpretation of crystallisation images can exceed the accuracy of human interpretation of the images [20]. This work (Machine Recognition of Crystallization Outcomes or "MARCO") required the compilation of a large training set of classified images, which were provided by a consortium of academic, government and industrial structural biology laboratories. As there are no standard classification schemes for describing a crystallisation trial [21], or an image of the trial droplet, each of the individual structural biology laboratories that contributed images used their own local classification system, which were collapsed down to four classes—'Crystal', 'Precipitate', 'Clear' and 'Other'. The four classes were chosen as they cover much of the gross information in a crystallisation droplet. Although the most interesting result for a crystallographer is 'Crystal', it is also the most rare; a paper from a decade ago estimated that harvestable crystals (*i.e.* those gauged by visual inspection to be potentially useful for diffraction experiments) are a small subset of all the crystals seen and make up 1–2% of the trial outcomes [22]. The classifications 'Clear' and 'Precipitate' help to identify trials which did not reach supersaturation and/or did not support nucleation ('Clear') or became over-supersaturated or disrupted the protein

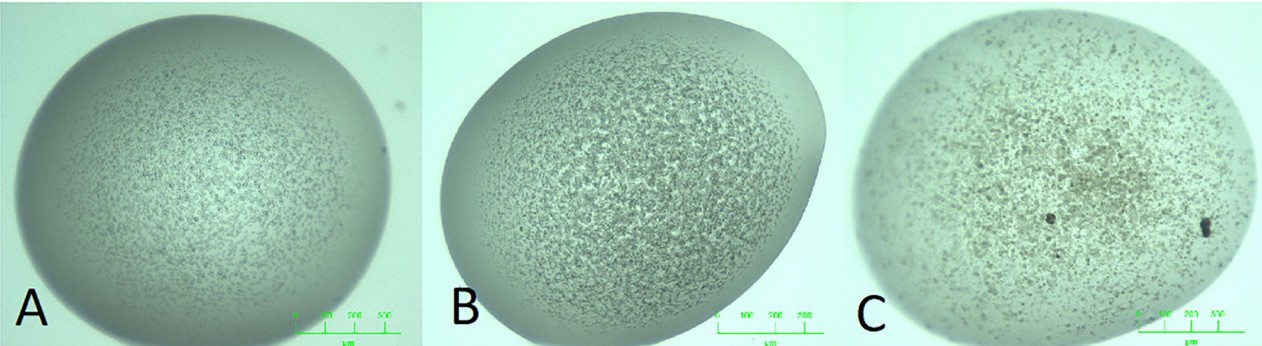

**Fig 1. Blurred boundaries between MARCO classes.** The three images here look similar, but have been classifed into three different MARCO classes by a crystallographer A: Other (phase separation). B: Precipitate. C. Crystal.

sample integrity ('Precipitate'). These non-crystal results can help locate an appropriate region of chemical space for further trials. The category 'Other' applies to all results that do not fall into the other three categories, and includes badly set up and mis-imaged experiments as well as trial outcomes like liquid-liquid phase separation and spherulites.

The outcomes from crystallisation trials are complex, and the classification of images of the trials into four classes is a gross simplification of the outcome space. The boundaries between any classification are blurred [23]—for example, very small crystals (microcrystals) can be indistinguishable from precipitate, and the human interpretation of outcomes can be variable, see Fig 1. Many drops display more than one outcome—for example, a drop might contain both crystals and precipitate. Drops may also display outcomes not captured by any of the classes—e.g. wrinkled protein skin, container imperfections or fungal growth, Fig 2. Futher, humans also focus on the domain relevant result: an image that contains a droplet with a dust fibre in it may well be classified as 'Clear' as the dust is considered irrelevant to the crystallisation experiment.

Any ML application is dependent on the quality of the data used to train the system—the MARCO model used half a million images from 5 different groups, and gives impressive results [24]. However, during the process of training the MARCO model it was noticed that the model's performance on images from our lab (The Collaborative Crystallisation Centre

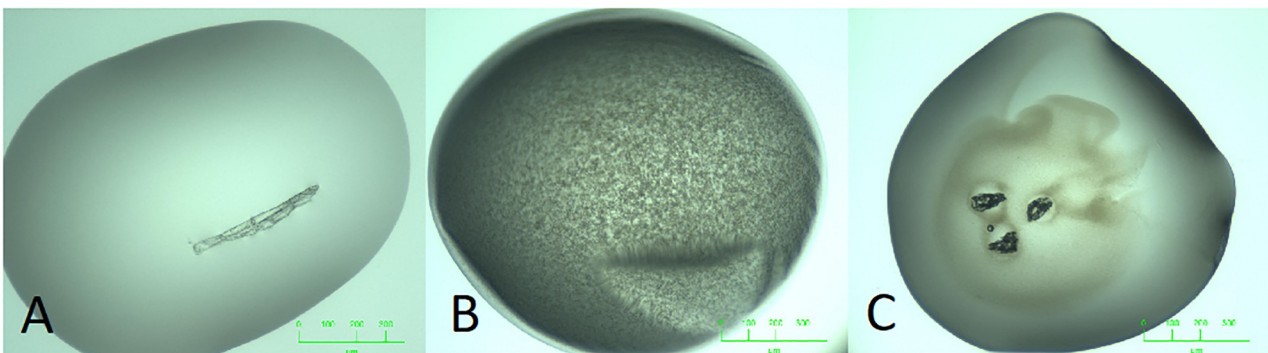

**Fig 2. Outcomes not captured in the MARCO closed set.** Many crystallisation images display outcomes that are not easily captured in four classes A: Dust fibre in an otherwise clear drop. B: Precipitated protein, along with an artifact from a wrinkled skin over the droplet. C. Collapsed air bubbles, (introduced inadvertently during drop dispensing) against a background of precipitated protein.

(C3) at CSIRO in Parkville, Australia) improved dramatically when a small number of images from C3 were included in the training set. This is a truism in ML; including local data into a general model invariably improves local performance [25]. Less obvious is how much local data is needed [26], and what quality [diversity, accuracy] of the local data is needed [27]. Another question is how best to utilise local data: *via* incorporation into an existing model (transfer learning [28]), or by building a new model using both the general data (here the MARCO training set) and local data [29]. Modern ML tools also allow for the re-examination of the approach of using local data only [30]. We were interested in understanding the dataset used to create the original ML model. Can we develop tools to measure the quality of these data? For example, we would like to be able to measure how much of the MARCO data is unique, and how representative of all crystallisation data the MARCO training data are.

In this current work we have contributed by:

- Characterising and identifying problems with the MARCO data set.

- Defining the parameters that allow the creation of models using currently available architectures that exceed the success of the original Inception V3 model.

- Developing guidelines for which local images to include to most effectively train a ML model using both MARCO and local images with the goal of producing an image classification model tuned to that local environment.

- Making code available to enable the facile development of a novel classification model tuned to the local environment which which uses both the MARCO training set and a limited amount of local data. (See DOI 10.5281/zenodo.7647927 or https://github.com/nrosa/MARCO-local).

- Creating a freely available protein crystallization dataset that contains both visible and ultraviolet light images. (The C3 Test dataset is available at https://zenodo.org/record/4635300).

## 2. Approach

### 2.1 Creation of a C3 validation set

The fundamental goal of this work was to investigate how the MARCO model and dataset can be effectively adapted to a local environment. In order to evaluate methods we developed a validation dataset. This dataset (C3 Test) uses the same meta-labels that were chosen for the original MARCO dataset: Clear, Crystals, Precipitate and Other. All of the images in C3 test are of vapour diffusion experiments, set up in Innovadyne SD-2 plastic plates and imaged on a Formulatrix RockImager; thus similar to many of the images in the MARCO dataset—by eye, the vapour diffusion images in the C3 Test set and the vapour diffusion images from the MARCO training set are indistinguishable. To ensure that there were no obviously redundant images in the C3 Test set only one image from any image timecourse was included.

### 2.2 Replicating the MARCO results

The model produced in the original MARCO paper used an InceptionV3 architecture [19]. Although this architecture achieved state of the art results upon its release in 2015 (https://paperswithcode.com/sota/image-classification-on-imagenet), the machine learning community has since transitioned to using ResNet architectures as the default [31]. In order to align our work with the community we trained models using architectures from the ResNet family with the aim of replicating the performance of the original MARCO model.

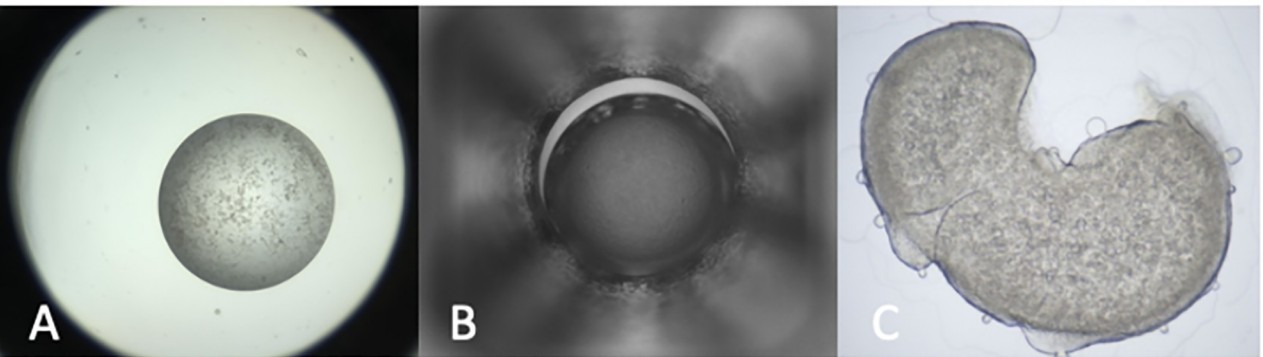

**Fig 3. Different experimental setups found in the MARCO training set.** Images of three different crystallisation trial types are included in the MARCO dataset image A: Vapor Diffusion (the most abundant), images of these experiments were contributed by GSK, MERCK, C3 and BMS. B: Microbatch under oil. Most of the images from this type of experiment were from the HWI, with a small number from GSK. C. LCP bolus—there are few of these in the dataset, and were provided by MERCK.

## 2.3 Image types in the MARCO dataset

During the visual inspection of the MARCO dataset we observed at least three different experimental setups [9]: under oil, vapour diffusion and a minor subset of Lipidic Cubic Phase (LCP) experiments, a type of liquid diffusion trial in which the concentrated protein sample is pre-mixed with a lipid such as monoolein [32], see Fig 3. To a crystallographer, images of LCP trials appear quite different from images of the other two types of crystallization experiment. Further, LCP experiments also show a greater range of outcomes than aqueous droplet trials as changes in the lipid phase contribute to image complexity along with the more commonly observed protein outcomes. There is no information about the number of LCP images in the MARCO dataset; it is not clear that the authors of the original MARCO work appreciated that any LCP images had been included in the dataset.

We estimated the number of LCP images by training a mobileNetV2 [33] architecture as a simple LCP classifier. Further, we examined if the number of LCP images in the MARCO dataset would be enough to produce robust classification of these type of images, using a small number of weakly labelled LCP images from the C3 laboratory.

## 2.4 Redundancy of images in the MARCO training set

By observing the change in classification performance on the MARCO test data as increasing numbers of images are removed from the training data we can estimate the redundancy of the dataset. We used the approach of Birodkar [34] to estimate the redundancy of the dataset by removing semantically duplicate images in a stepwise manner. Semantic duplicates were determined by performing agglomerative clustering [35], in which images with the highest similarity are grouped together into clusters. We use the euclidean distance between images in the representation space of a ResNet50 model [31] that was trained using the entire MARCO dataset as our measure of similarity. The image that was closest to the centre of the cluster was retained in the dataset whilst all others were removed. The number of clusters formed was chosen to be the number of images that were to be retained in the reduced dataset; thus to remove 10% of images in a hypothetical dataset containing 100 images, 90 clusters would be chosen. We made one modification to the published method whereby we performed the agglomerative clustering within pseudo-classes that were created by K-means clustering [36] within each true

class within the representation space. Each pseudo-class contained approximately 10K images. This change was made to allow the analysis to run within a reasonable amount of time.

## 2.5 Image and dataset characterisation

To characterize and understand the MARCO training set, we used dimension reduction to probe for any inherent structure within the MARCO training data.

A model was trained to extract features from input images that were useful for differentiating between semantically different images, without imposing bias from our preconceived notion of outcome class, using a Simple framework for Contrastive Learning of visual Representations (SimCLR) [37]. SimCLR is an algorithm for training a feature extractor in an self-supervised (*i.e.* label free) manner.

Uniform Manifold Approximation and Projection (UMAP) [38] was used to reduce the high dimensional representations produced by SimCLR to two dimensions in order to visualise the MARCO test dataset. The reduced dimensional spaces learned by UMAP generally provide a better view of the global structure of a dataset than alternative approaches that seek to preserve local topological structure. We also used UMAP to visualize MARCO test data representations generated from an intermediate layer in the MARCO model.

To understand the distribution of sample positions relative to class boundaries we used the methods described by Liu *et al* [39] to find the six binary class boundaries between each class pair; then we plotted the distribution of distances between each boundary and each sample in the MARCO and C3 test sets.

## 2.6 Adding local images to the MARCO training set

We explored different training setups where the original MARCO training dataset was supplemented with local C3 images and observed what parameters created the model that generalized best to the C3 setup. The two variables we tested were: image class distribution, and the number of local images included. To obtain images that could be added to the MARCO dataset we searched C3's collection for images that had been scored by an expert. We used the same criteria that were used in section Creation of a C3 validation set to select images, however care was taken to ensure that there was no overlap between the validation dataset (the C3 test set) and this image pool. The score from the single crystallographer was taken as ground truth.

We also explored using both entropy and cross entropy as 'hardness' measures to select local images for inclusion into the MARCO training data. 'Hard' local images were found by using the entropy of model prediction as a proxy for hardness and selecting images from the pool which had the largest entropy values for any given model. We compared models where hard local images were included to models with the same number and distribution of randomly selected images were included.

## 3. Experiments

### 3.1 Creation of a C3 validation set

To find images, C3's collection of experiments from January 2017 onwards was scraped for images that had already been labelled by at least one crystallographer (who was not in the relabelling team, see below), this gave us a starting set of 9215 labelled images. Since labelling crystallization outcomes is known to be a hard problem, even for experts (see for example [40]), we increased the robustness of our labels with a two-step re-labelling procedure. In the first round of re-labelling all images were labelled by a second crystallographer. A label was only considered robust if it was chosen by both the two crystallographers (a majority score). In the

**Table 1. Label distribution of different data sources.**

| Data source | Clear | Precipitate | Crystal | Other |
|---|---|---|---|---|
| MARCO | 33.6% | 48% | 12.8% | 5.6% |
| C3 Test | 23.5% | 40.0% | 28.3% | 8.1% |
| Cinder | 27% | 50% | 11% | 12% |

Class distributions found in the MARCO data and the C3 test data. These can be compared to the distributions found from the Cinder application [21], which may be an unbiased indication of the native distribution of outcomes.

second re-labelling round all images that were yet to have a consistent label were scored again by third crystallographer. In neither the two rounds of relabelling were the existing labels revealed, so every score was assigned by simply observing the image. After the second round of additional scoring, 347 of the original 9215 images remained ambiguous with each crystallographer choosing a different score; these images were not included in the final dataset. The breakdown of the scoring for each of the two validation rounds can be seen in S2 Appendix. The class breakdown of the final C3 Test dataset is shown in Table 1.

Both protein crystals and crystals of inorganic salts can grow in protein crystallisation trials. Images taken with ultra-violet (UV) light may be useful to distinguish between salt and protein crystals, as [most] protein crystals glow under UV illumination due to the intrinsic tryptophan fluorescence of protein crystals [41]. UV imaging is quite widely applied in crystallization laboratories; and the orthogonal information these images provide could be helpful for classifying crystallization experiment outcomes. Generally, a UV image of a crystallization experiment is taken at the same time as a visible light image is collected, and we have included the associated UV image in the C3 Test dataset. To the best of our knowledge this is the first publicly available crystallization outcome dataset to include labelled UV light images.

## 3.2 Replicating the MARCO results

In order to replicate the performance of the MARCO model, we did a large exploration of training hyper-parameters using two sizes of the Resnet architecture, Resnet18 and Resnet50, along with multiple image resolutions. More information about the training procedure used can be found in the S1 Appendix. We compared the accuracy of trained models using both the MARCO test set and the C3 Test set. The best outcomes of our experiments are shown in Table 2. Due to the variance of individual training runs the reported accuracy for each model

**Table 2. Performance of different ResNet models on the MARCO and C3 test datasets.**

| Model | Image size | MARCO Acc. | C3 Acc. |
|---|---|---|---|
| MARCO | 599 | 94.5* | 82.03 |
| Resnet18 | 256 | 93.21 | 79.51 |
| Resnet18◇ | 512 | 94.18 | 82.12 |
| Resnet18◇ | 600 | 94.21 | **82.14** |
| Resnet50 | 256 | 94.24 | 76.46 |
| Resnet50 | 512 | **94.63** | 81.5 |

The evaluation performance of several models trained upon the MARCO dataset. Best results are in **bold**. Results marked * were copied directly from [20]. Models marked ◇ were trained for 50 epochs. The reported accuracy for each model is the average of three trials using different random seeds. More information about the training procedure used for these models can be found in the S1 Appendix.

is the average of three trials which were run using three different random seeds. We found high-resolution Resnet18 models were less prone to over-fitting on the MARCO data distribution so could be trained for longer, therefore we trained these models for 50 epochs rather than the 30 epochs used for the Resnet50 models.

### 3.3 Image types in MARCO

There are images of three different types of crystallization experiments in the MARCO data. Most common are images of vapor diffusion experiments; all the contributing groups—with the exception of HWI—provided images of this type of experiment. HWI provided images of crystallization experiments which had been set up under a column of oil (microbatch experiments). Finally, some groups (neither CSIRO nor HWI) also included some images showing Lipidic Cubic Phase (LCP) experiments. Although the outcomes of the three different experiment types are broadly similar, the images themselves look significantly different, and LCP experiments can show features not seen in the other experiments. We created a binary classifier using mobileNetV2, training the model with 1000 LCP images from CSIRO C3 and the same number and distribution of non-LCP images selected from the MARCO image dataset—the 1000 MARCO images were checked by hand to ensure that these images were of vapour diffusion or microbatch experiments. This model was used to scan the MARCO training data, and identified 26K images which were classified as putative images of LCP experiments; human inspection showed about 1 in 10 of the 26K images thus identified were actually LCP images; we estimate there are on the order of 2.5–3K LCP images in the entire MARCO dataset. A test of how well the MARCO model works with LCP images was done by selecting 960 labelled LCP images from C3, and calculating the confusion matrix generated from the MARCO scoring of this small set of images, see Fig 4.

### 3.4 Redundancy of images in the MARCO training set

The accuracy on both the C3 Test set and the MARCO validation set was calculated using a Resnet50 model with images of $256 \times 256$ pixels for 5 subsets of the MARCO training set, with increasingly stringent requirements for distinctness of the images (i.e. with decreasing redundancy, as per the method of Brodikar, described above). Models were trained in triplicate, with the reported accuracies being the average across the three runs. Even with 50% of the original data, less than 0.3% decrease in accuracy on the MARCO test data was observed, Fig 5. Interestingly, the accuracy of the classification of the both MARCO and C3 test data peaked when using 80% of the MARCO training data, suggesting that the very redundant nature of the original MARCO data leads to a degree of overfitting on the redundant data at the expense of accuracy. The reduced datasets used in this analysis can be found in the code repository.

### 3.5 Image and dataset characterisation

UMAP was run using the penultimate layer of a ResNet50 trained from a random initialisation using the MARCO dataset to produce the plot shown in Fig 6. Coloring the points by either class or by source, we can see a stronger clustering based on the MARCO classes, this makes sense as MARCO was trained to differentiate images based on these classes.

By training a SimCLR model with a ResNet50 backbone, using the unsupervised contrastive learning approach, we produced the plots in Fig 7. The points were coloured by either class or by source and one observes a strong clustering based on the images' origin. That is, the dominant feature SimCLR observes during a self-supervised training is the image source. A more granular breakdown of these plots can be found in supplementary information.

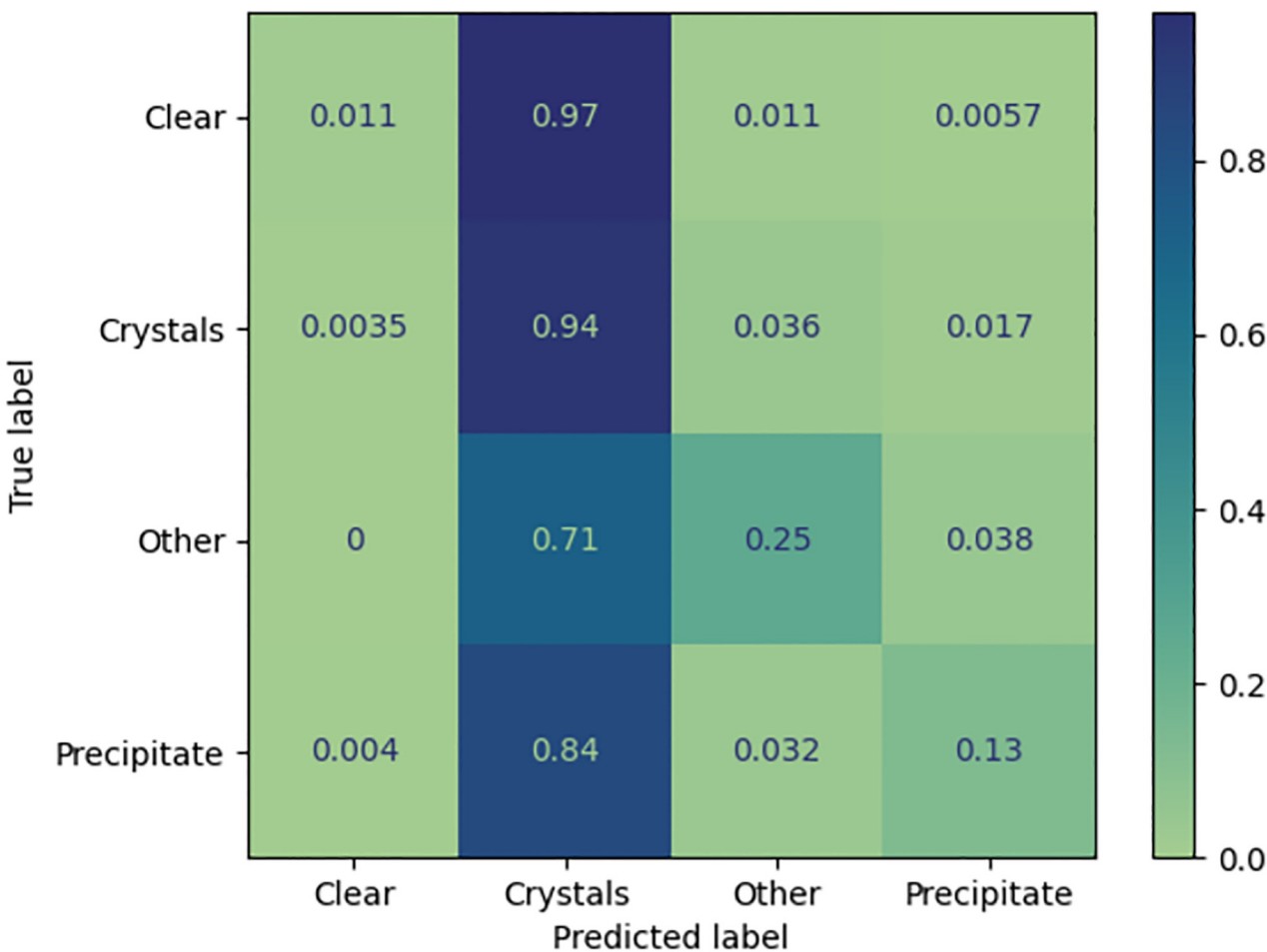

**Fig 4. Confusion matrix for 960 LCP images from C3, classified using the MARCO model.** The confusion matrix is normalized over the true conditions (e.g. rows), this shows that the MARCO model classifies most LCP images as crystals, overall accuracy 56%.

We used the weights from a Resnet50 trained upon the MARCO dataset and used the techniques in [39] to find the decision boundaries for each combination of pairs of classes. Then for each sample in both the C3 and MARCO test sets we find the distance to each decision boundary, see Fig 8.

### 3.6 Adding local images to the MARCO training set

We evaluated the local data performance with the addition of weakly labelled local data. For these experiments a Resnet50 was used with images of 256 resolution. The results are shown in Fig 9.

## 4. Discussion

The early attempts to develop machine learning tools to interpret crystallisation images failed to come close to human scoring accuracy for (at least) two reasons: firstly, by not having access to a sufficiently large, diverse and well-classified training set and secondly, many of the attempts predated the breakthrough applications of Convolutional Neural Networks (CNNs) in 2012 [42]. The MARCO consortium brought together images from five different institutes,

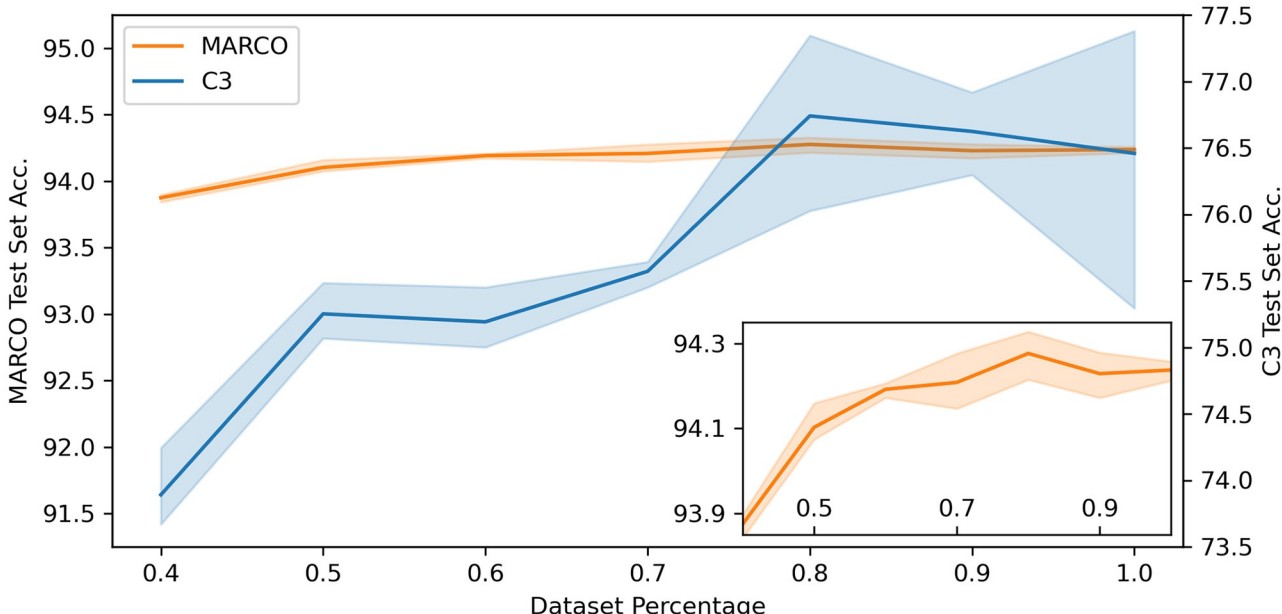

**Fig 5. Performance of a set of Resnet50 models using 256 × 256 pixel images on the C3 Test dataset and MARCO validation dataset when trained with increasingly strict redundancy cuts.** The blue line shows the accuracy of the reduced data models against the C3 Test data, and the orange line shows the accurancy of the reduced data models against the MARCO validation set. 10% of the data were removed stepwise from 100% to 40% of the original training data. The pale regions cover the area defined by the standard deviation of three runs. Insert shows detail of the MARCO validation data trials.

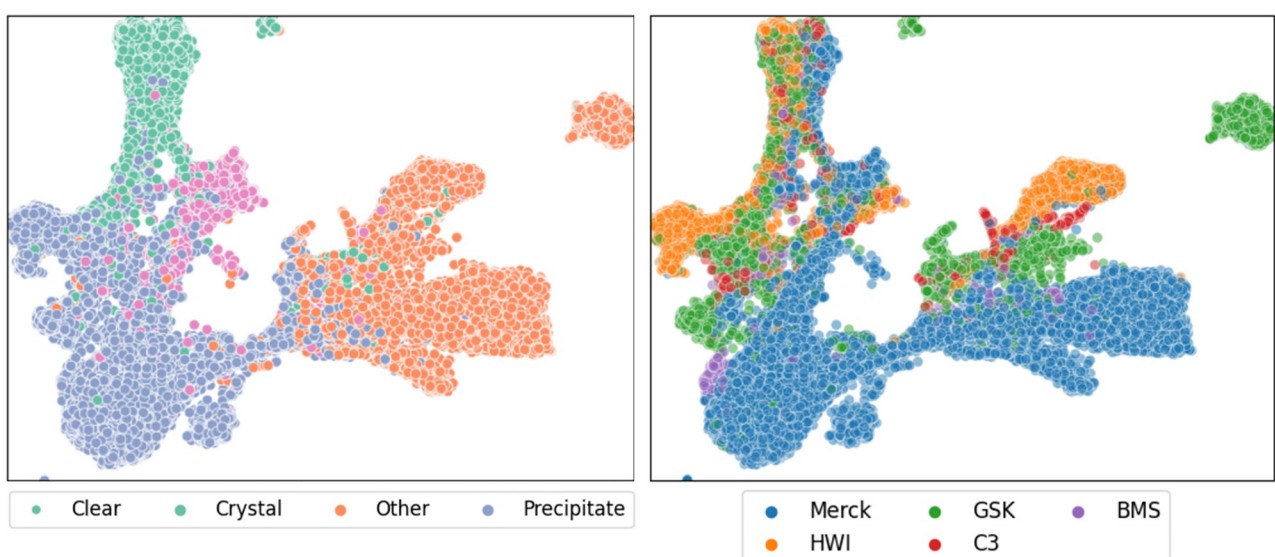

**Fig 6. UMAP projection of labelled data from the MARCO validation dataset.** The left hand figure shows the projection from the ResNet 50 model coloured by classification, where clear, crystal, other and precipitate are coloured green, pink, blue and orange respectively. The right hand figure shows the same projection, but coloured by source of the images—Merck, GSK, BMS, HWI and C3 are coloured blue, green, purple, orange and red respectively. A similar UMAP projection using the original Inception V3 model from 2017 is shown in S1 Fig in S3 Appendix.

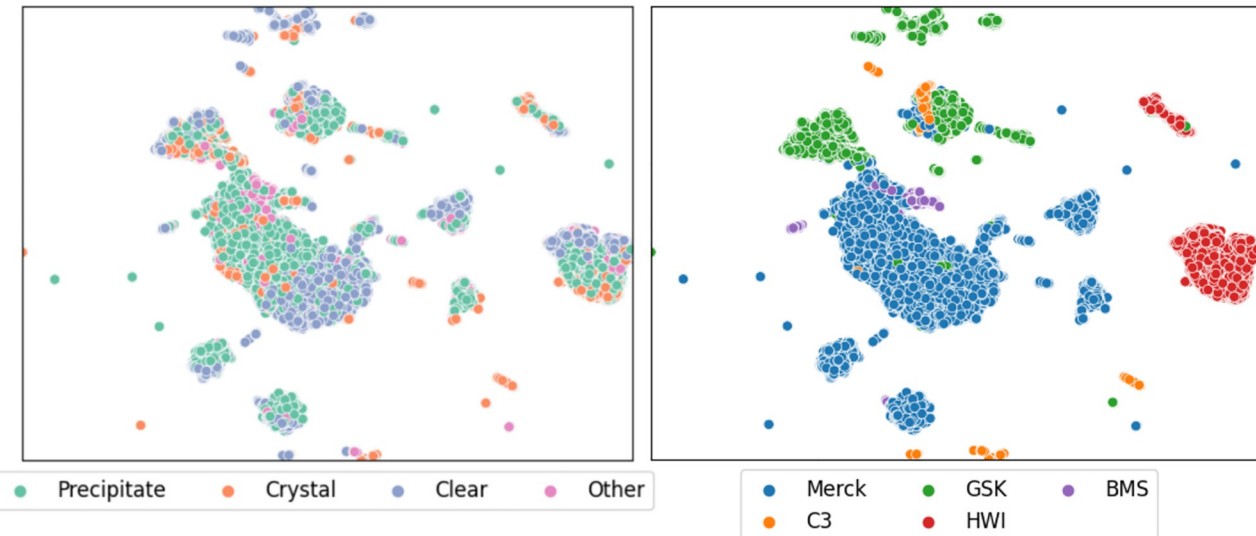

**Fig 7. UMAP projection of data from the MARCO validation dataset labelled using a SimCLR unsupervised approach.** The left hand figure shows the projection from the ResNet 50 model coloured by classification, where Precipitate, Crystal, Clear and Other are coloured green, orange, blue and pink respectively. The right hand figure shows the same projection, but coloured by source of the images—Merck, GSK, BMS, HWI and C3 are coloured blue, green, purple, orange and red respectively. A similar UMAP projection using the original Inception V3 model from 2017 is shown in S2 Fig in S3 Appendix.

creating a larger and more diverse dataset than had been used before for training crystallisation image classification models, and also tapped into the established expertise in CNNs. The Inception V3 model produced from the MARCO dataset gave impressive results against a test set of data culled from the MARCO data, however, the performance of the model drops significantly when tested against local data.

Approximately 423K images of varying image sizes make up the MARCO training set. These images are either 5 megapixel (MP) images obtained from commercial imagers or 2 MP images collected from the custom built imagers at the HWI [43]. The images were labelled with one of four possible labels, 'Crystal', 'Clear', 'Precipitate', 'Other'. The images were initially classified using local classification systems, which totaled 38 different labels. These local categories were mapped to the four meta-classes described above during the collation of the dataset. Subtleties in classification which may have been captured with local classification systems were lost during the relabelling; for example, at CSIRO C3, there are six classes relating to 'Crystal' outcomes (Crystalline, Crystal*, Crystal**, Crystal***, 'shoot me', possible salt crystal, confirmed salt crystal) which all ended up with the same 'Crystal' classification. We were surprised to observe a small number of LCP images in the original MARCO data. Using a binary classification system we estimate that images of LCP droplets make up less than 1% of the images.

Although the MARCO training dataset is large there are still limitations to the both the accuracy of the labels and to the diversity of the data. For example, of the five groups that contributed images to the training set, four used the same (or similar) type of crystallisation plate to set up the trial droplets—one of the five sitting drop plates manufactured by Swissci UK. There are many other crystallisation plate types on the market (see, for example, https://hamptonresearch.com/cat-187.html), each will have different imaging properties. For trials set up on these different plates to be classified well, the classification model would need to be robust to a domain change or image distribution shift. Interestingly, despite C3 currently

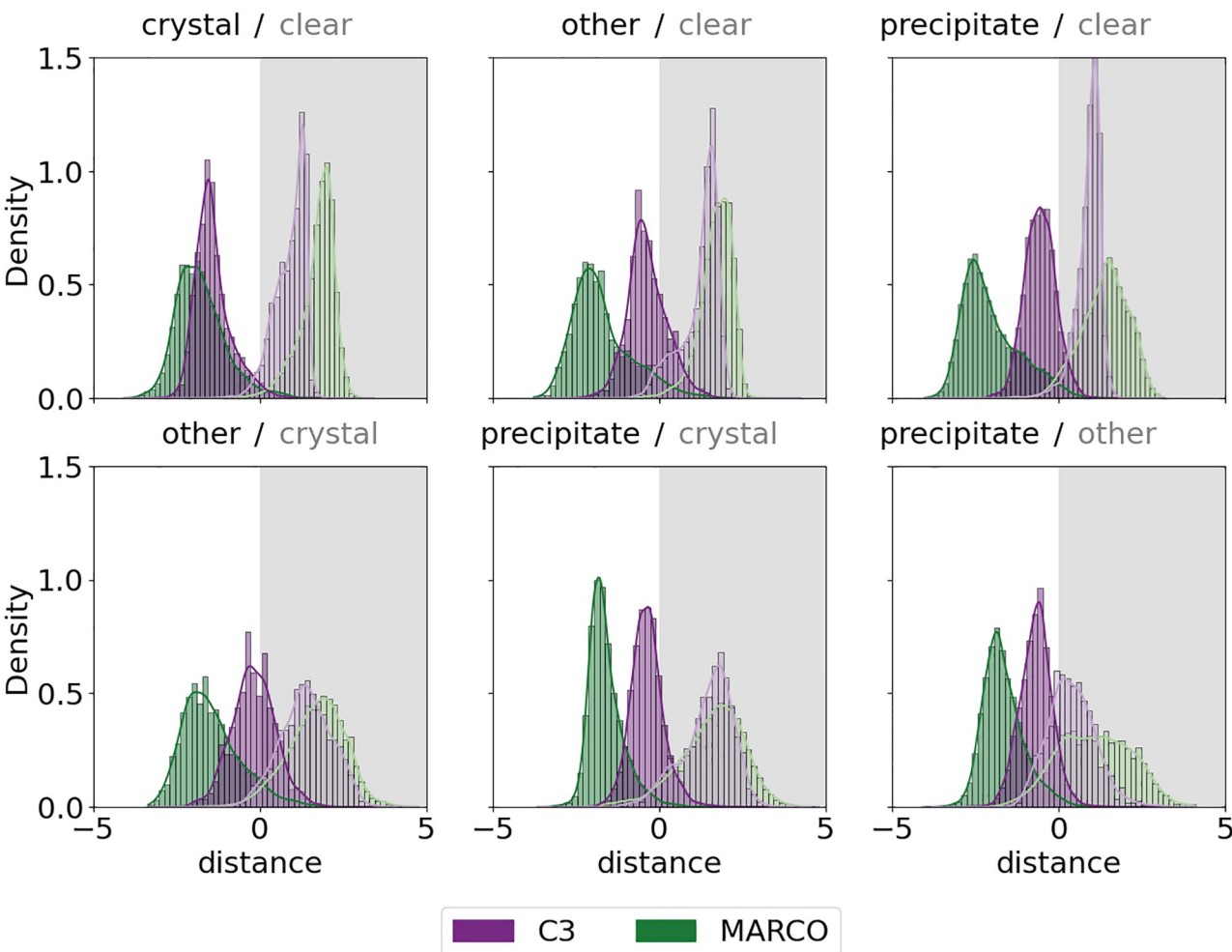

**Fig 8. Histograms of distances of images from the decision boundaries.** Images from the C3 test set are shown in purple, and those from the MARCO validation set are shown in green. The four classes give 6 decision boundaries, which are shown as clear/grey boxes. The C3 images are closer to the decision boundaries as defined by the MARCO training set, and are are not symmetric around the decision boundaries.

using the same Swissci plate type and same Formulatrix imaging system as were used by at least two groups in the original MARCO data, there is a performance reduction for these C3 images. This suggests there are further factors affecting the domain beyond the obvious plate and imager types. Although C3 images were included in the original MARCO data, those images were produced from older equipment, supplied by a different vendor. The more recent C3 images, that were used for the creation of the C3 test set, are not represented in the original MARCO data, and thus mimic the situation found in most laboratories (that is, those which were not part of the MARCO consortium).

Within each organisation, more than one set of automation was used to create the images in the MARCO dataset—both the equipment used to set up crystallization droplets and to image them varied within each organisation. For example, the images that C3 contributed to the original MARCO dataset were collected on two different imaging systems, and the drops themselves were set up with three different dispensing robots. The images from Merck were collected on 6 different machines, and the images from GlaxoSmithKline also used 6 different imaging setups.

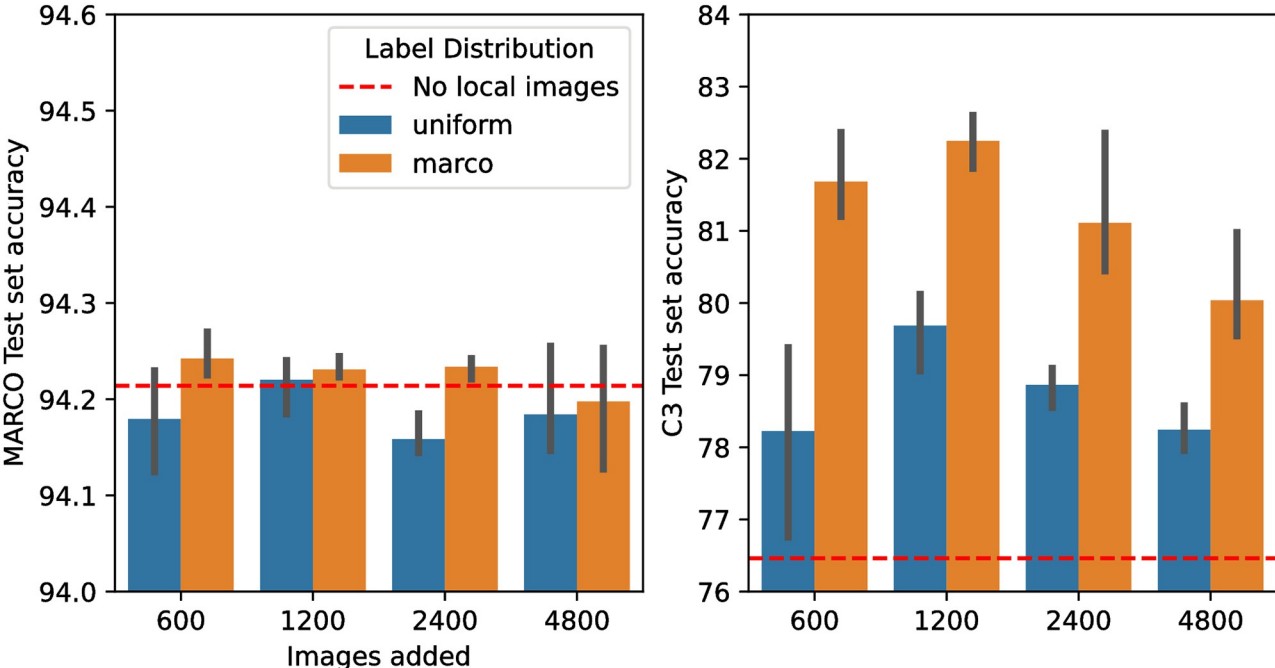

**Fig 9.** Performance of Resnet50 architecture using 256 × 256 pixel images on the MARCO validation dataset (Left hand graph) and C3 dataset (Right hand graph) when supplementing the training set with different numbers of images from a weakly labelled pool of local images. Trials varied both the number of images included and the class distribution of the included images—blue bars have a uniform class distribution and orange bars show the results when the class distribution of the included images matched the distribution of classes seen in the original MARCO training set (see Table 1). Error bars are the standard deviation of three runs. The dashed red line shows the accuracy of the unmodified training set for comparison. The scales on the two graphs are different: little change in the performance against the MARCO training set is seen irrespective of the number of added images; the accuracy of the model trained with MARCO data supplemented with 1200 local images increases by 6%.

We suspect that much of the data used to train the MARCO model were not initially compiled with the goal of creating a training set, but were scraped from images that had already been scored for other reasons. If so, it will certainly have affected the quality of the MARCO data. Firstly, the re-purposed images may not be perfectly classified, as creating a "ground truth" dataset was not the goal of the original scoring process. Ensuring accurate classification is difficult, particularly given the range of outcomes seen in a crystallisation trial: experience looking at crystallisation trials is essential for correctly interpreting outcomes. It might be that interns or other early career researchers were hired to classify large numbers of images so that more senior crystallographers could be presented with a set of images which were enriched for promising results. In this case the classifications would most likely be biased towards over-estimating positive results, to ensure that any promising image was retained for further examination. Another consequence might be that less effort would have been put into classifying the non-crystal containing images correctly; thus non-crystal images may be more weakly labelled than crystal images, or may simply be underrepresented in the labelled data. Further, there would have been no reason to ensure that images of the same drop (collected as part of a time course) were excluded from the dataset. Fig 10 shows examples of images from timecourses that appear in the dataset. We have shown that the MARCO data is highly redundant, adding weight to the idea of MARCO dataset being composed of (at least some) repurposed images. Including many images from a single timecourse of the same droplet will increase the absolute size of the training set; however, because these images are likely near duplicates of each other they can lead to bias in the measurement of the model's performance [44]. Furthermore, the

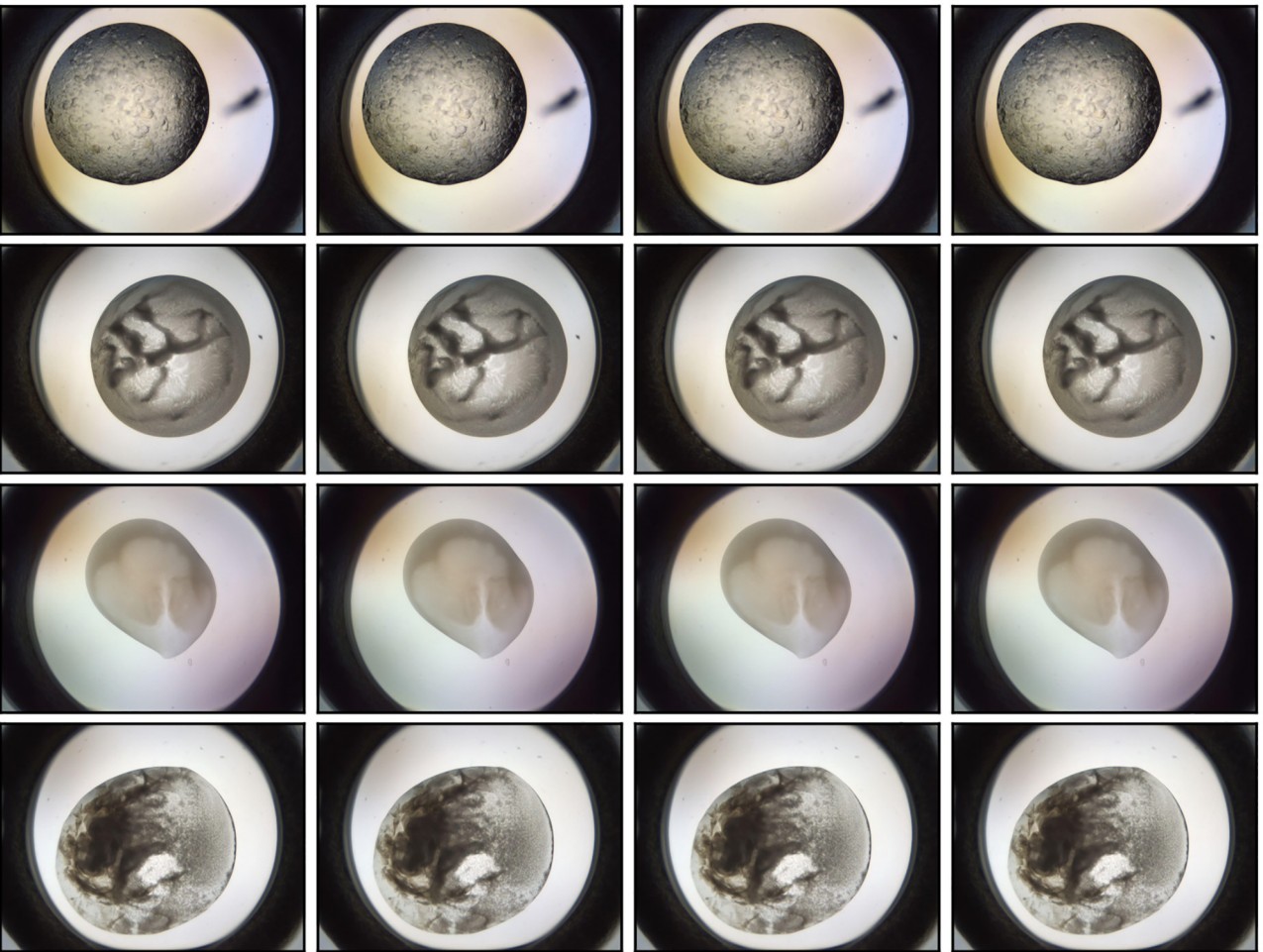

**Fig 10. Examples of redundant images contained within the MARCO dataset.** Each of these 16 images is found in the MARCO dataset, and are clustered into rows based on similarity as determined by the semantic redundancy process described in this Section. The similar images shown here are different timepoints of the same crystallisation experiment. Each image is so similar they cannot be distinguished apart by a human. There are other semantically redundant images that don't appear the same, but contribute the same information to the model (not shown).

MARCO data were split into a training set and a validation (test) set; given the high level of redundancy of the data the training and test sets would almost certainly not be independent, thus the stated performance of the model in the work published in 2018 [20] will have been overestimated. We show that excluding the 20% most redundant images made no difference to the accuracy of the model, and only a small drop in model performance was observed when 60% of the most redundant images were excluded. Redundant images can be images with similar information content, even if they are not simply different timepoints of the same droplet.

Much of the outcome space of a crystallisation trial is a function of the protein sample in the trial, rather than the chemical cocktail used. Thus if the same protein is set up against many different cocktails, the same protein-dependent set of outcomes is likely, so that the diversity of the image data depends not only on the number of trials, but the number of different samples used to generate those trials. Industrial protein crystallographers are most likely to work in large pharmaceutical companies, where they focus on developing SAR (Structure Activity Relationship) for proteins which are targets for pharmaceutical intervention. This process requires that a limited number of proteins are subjected to extensive crystallisation trials,

in turn limiting the possible outcome space. Ideally the images collected for a MARCO type dataset should come from many different laboratories, so to have as broad a sampling of crystallisation outcome space as possible.

The MARCO dataset contains a small (2.5–3K) number of images capturing LCP setups, which would certainly have increased the diversity of the dataset, as LCP experiments consist of a bolus of lipid/protein mesophase surrounded by a small volume of precipitant. The possible outcomes from the LCP trials include breakdown of the mesophase or transformation of the phase [32] as well as the outcomes seen in other crystallisation droplets. However, LCP experiments may be sufficiently divergent from more traditional crystallisation trials as to not expand the outcome space in a useful manner. The MARCO model did not reliably classify LCP images from the C3 laboratory, see Fig 4. This result needs to be tempered by understanding that there were no C3 generated LCP images in the MARCO dataset, and this current work highlights how important including local data in the training set is to the accuracy of classification of local images. However, it could well be that the small number of LCP images in the MARCO data simply do no provide enough coverage of possible LCP outcome space to ever provide accurate classification of LCP images. Further, it could be that the simplistic 4 class classification scheme which only crudely covers soluble crystallisation trial outcomes is not granular enough to capture the results seen in LCP experiments.

To get a more tactile representation of the MARCO data, we used a UMAP projection to visualise the MARCO test dataset. UMAP reduces the high-dimensional vector that MARCO uses to calculate probabilities into a two dimensional point we can visualise in a scatter plot. This approach allows us to identify natural structure that occurs in the MARCO data as seen by the MARCO model. Naturally, the representations taken from intermediate MARCO layers will be coerced by the supervised learning process of the MARCO model. To overcome this, a model was trained using SimCLR, a self-supervised method. SimCLR encourages a model to learn representations it can use to differentiate between images of different things. We repeated the dimension reduction approach using the self-supervised model representations and observed the structures upon which the model learned to focus. As shown in section Image and dataset characterisation the largest axis of variation in crystallisation images can be attributed the source laboratory. This indicated a strong difference between images from different laboratories.

When using an existing model to classify local data there is a domain shift [45], which causes a drop in performance. One way to circumvent this issue is to retrain the model with images from the new source included in the training data; this is the approach taken here. The local data used for retraining were scored (classified) by a single person. As classification of crystallisation images is a difficult problem—both this current work and previous work [23] show that single-person scoring is not robust—we refer to these local images as "weakly labelled". Weakly labelled images were used as these are most readily available in crystallography laboratories, as the procedure for producing a robustly labelled local dataset is expensive in both time and human resources. In order to provide a retraining protocol that could be adopted by any crystallisation laboratory, we tested different combinations of parameters using recent data from the C3 laboratory. As the more recently acquired C3 images were not included in the MARCO dataset, this mimics the situation found in most crystallisation laboratories. Our results suggest that whilst model capacity (number of parameters) is important for performance on the MARCO test set, it does not help with generalization and may even harm performance at lower image resolutions. The most important factor for inclusion of local data from a laboratory—as suggested by our retraining results—is image resolution. This supports the assumption presented by Bruno *et al.* [20], that as crystallisation involves detailed, thin structures a larger input image may yield better outcomes. We hypothesize resolution has

a smaller impact on the MARCO test set results because images from the MARCO test set share the same data distribution as the training set. During training the model can use the ground truth information to learn to hallucinate (imagine) detailed features from the low resolution image. This learnt relationship still applies to the MARCO test set because the MARCO test and training data sets have the same data distribution. However, because the data distribution is most likely different between the C3 test set and the MARCO data the learnt relationship between low resolution images and the detailed features in the corresponding high-resolution images is no longer valid.

When selecting images to supplement the original MARCO images, we firstly considered the class distribution as it is known to be an important variable [46]. A naïve choice is to randomly select images from a large pool, however, this would likely lead to a biased class distribution which could cause the model to favour any class which is sampled more in the added images. Other options are to match the the label distribution found in the original MARCO dataset; to use a uniform distribution to equally condition each class to the new source images or to match the label distribution of the local images. In all cases, adding images with a label distribution that matches the original MARCO dataset outperforms adding an equal amount for each class, even when evaluated on the C3 test set which, as shown in Table 1, has a different class distribution. The MARCO test set performance slowly decreases as the number of local images are added which is as expected since the model is using capacity to learn about the local image type. Surprisingly the C3 test set performance increased when 600 and 1200 local images were added but then began to decay when more images were added. This is counter-intuitive as one would expect that adding further local images would continue to increase performance. We conjecture that this decay is due to the weakly labelled nature of the image pool. With a small number of images the model is conditioned to the local images. However, as the number of images increases the poor labels start to negatively impact the classification performance.

The label distribution seen in any laboratory does not necessarily reflect the underlying class distribution, as—most often—not every image is labelled. A human with limited time will tend to label only the more domain-interesting images ('Crystal'), and will often just skip labelling images if they fall into one of the other label classes. We estimate the true distribution of results (as opposed to the distribution of labels) using data from the Cinder mobile application [21]. In this app, images were presented to the viewer one by one, and the viewer uses the app to associate one of four labels to the image. The user can skip labelling an image, but there is a time penalty for doing so. We believe that the labelling statistics obtained from this app are possibly the best estimation available of the true (4-class) distribution in crystallisation. The MARCO and CINDER distributions are shown in Table 1. If we match the MARCO label distribution we will also approximately match the typical class distribution of images.

In other AI problems such as Object Detection, it has been shown that the process of *bootstrapping* or *hard negative mining* can improve performance [47]. Hard negative mining is a process in which examples that are 'easy' are removed from the training set and replaced by 'hard' examples. 'Easy' samples are ones which are correctly classified with a high confidence and 'hard' examples are ones which are incorrectly classified or classified with a low confidence. Such an image selection method could also benefit retraining to reduce performance drop for new images. Intuitively, this suggests that an 'easy' example is already within the original model's distribution—as it was well and confidently scored by the model, whereas a 'hard' example violates the original model and thus would teach the model more. We explored using both entropy and cross entropy as hardness measures to select local images for inclusion into the MARCO training data. However, neither improved model performance.

Another measure of hardness is distance from the decision boundary. Samples that are clearly distant from a class boundary are simple to classify, as they fall cleanly into a single class, but those which lie close to a decision boundary have some ambiguity about them, they must have some features that are similar to both classes. Fig 8 illustrates two features of the C3 test data as compared to the MARCO validation set. We consistently see the mean distance of the C3 Test set to the decision boundary is less than the distance from the decision boundary for the MARCO validation set. Furthermore, we can observe that a large fraction of the C3 test sample cross over the decision boundary, indicating that they are incorrectly classified by the reference model. These features are not only illustrative of the increased difficulty of the C3 dataset over the MARCO dataset for the MARCO model, but are also indicative of the distribution shift between datasets. The stark difference in distributions from C3 and MARCO in Fig 8 demonstrate the necessity to retrain or fine-tune the MARCO model when moving between labs to better align decision boundaries. The greater distance between the boundaries and the MARCO images could come from two sources—firstly, it could simply show that the validation set is not independent of the training set, which is likely given the redundancy of the data, but also it could support the notion that the MARCO dataset simply does not contain many ambiguous or difficult-to-classify images.

The UMAP projection of the unsupervised training using the MARCO data show that the axes of largest variation of crystallisation images in the SimCLR representation space are not equivalent to the labels associated by humans, but are dependant on the laboratory and equipment used to collect the images.

## 5. Conclusion

Although the MARCO model developed in 2018 has had a huge impact by producing reliable classifications of crystallisation images, it was believed that the success of the model was due to the extent of the training dataset. We have shown that the training data are redundant, and the true extent of the original MARCO data is probably less than 50% of the data that were used. The MARCO performance is significantly less impressive on local data than on test data derived from the same sources as the training data. We recommend including robustly labelled local data; further, our work suggests that there is an upper limit to the number of poorly labelled local images that can be included without decreasing model performance. For our local data we observe this value to be around 1000 local images.

## Supporting information

**S1 Appendix. Training procedure.** Complete breakdown of hyper-parameters used to train models.
(PDF)

**S2 Appendix. C3 dataset labelling.** S3 and S4 Tables show the correlation between labels after either one of two cycles of re-labelling.
(PDF)

**S3 Appendix. Projections of penultimate layer representations from the MARCO Inception V3 model and a SimCLR trained Resnet50.** S1, S2 Figs show similar data to Figs 6 and 7 respectively, further decomposed into each groundtruth label.
(PDF)

## Acknowledgments

We are grateful for the users of the Collaborative Crystallisation Centre of CSIRO for providing scores and samples used in this study. We thank Drs Alex Caputo, Matt Dennis, Bevan Marshall and Tom Peat for scoring the C3 test set.

## Author Contributions

**Conceptualization:** Janet Newman.

**Formal analysis:** Nicholas Rosa, Christopher J. Watkins.

**Investigation:** Nicholas Rosa, Christopher J. Watkins.

**Methodology:** Nicholas Rosa, Christopher J. Watkins.

**Project administration:** Janet Newman.

**Software:** Nicholas Rosa, Christopher J. Watkins.

**Supervision:** Janet Newman.

**Visualization:** Nicholas Rosa.

**Writing – original draft:** Nicholas Rosa, Christopher J. Watkins, Janet Newman.

**Writing – review & editing:** Nicholas Rosa, Christopher J. Watkins, Janet Newman.

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
