## [Decision Letter · Decision Letter 0]

5 Dec 2022

PONE-D-22-27063Moving beyond MARCOPLOS ONE

Dear Dr. Newman,

Thank you for submitting your manuscript to PLOS ONE. After careful consideration, we feel that it has merit but does not fully meet PLOS ONE’s publication criteria as it currently stands. Therefore, we invite you to submit a revised version of the manuscript that addresses the points raised during the review process.

We look forward to receiving your revised manuscript.

Kind regards,

Zhentian Wang, Ph.D.

Academic Editor

PLOS ONE

Journal Requirements:

Additional Editor Comments:

Please address the minor comments from the reviewers point by point and resubmit the manuscript.

Reviewers' comments:

Reviewer's Responses to Questions

**Comments to the Author**

1. Is the manuscript technically sound, and do the data support the conclusions?

Reviewer #1: Yes

Reviewer #2: Yes

Reviewer #3: Yes

2. Has the statistical analysis been performed appropriately and rigorously? 

Reviewer #1: Yes

Reviewer #2: Yes

Reviewer #3: Yes

3. Have the authors made all data underlying the findings in their manuscript fully available?

Reviewer #1: Yes

Reviewer #2: Yes

Reviewer #3: Yes

4. Is the manuscript presented in an intelligible fashion and written in standard English?

Reviewer #1: Yes

Reviewer #2: Yes

Reviewer #3: Yes

5. Review Comments to the Author

Reviewer #1: This thoughtful and careful analysis of ML for detecting crystallization sheds new light on the standard reference tool, MARCO, and offers a path forward toward future improvement. In short, this manuscript is quite interesting and should be published. A couple of small things should, however, be fixed.

1) p.11: a comment by the first author was left in the text (and unaddressed).

2) p. 18: the notion of hallucination of features should be better explained and/or referenced.

Reviewer #2: This paper is a welcome follow-up to the original MARCO study, which raised as many practical questions as it provided answers. The paper is well-written, and provides new insights to how the original findings from the MARCO study can be refined to become of practical use for a specific lab.

A few comments to the authors that are not commentary on the quality of the study, but could help direct further research:

1- While the ResNet class of models is a step up and a modernization of the original Inception-v3 model, it is by now also slightly outdated for many use-cases: modern Transformer-based architectures such as ViT (https://paperswithcode.com/paper/scaling-vision-transformers) and CoCa (https://paperswithcode.com/paper/coca-contrastive-captioners-are-image-text) have generally demonstrated superior performance *particularly* in a transfer learning setting. In other words, the world of computer vision is increasingly moving towards fine-tuning these (massive) models as opposed to training domain-specific models and seeing vastly improved transfer performance. Whether this finding transfers to the MARCO setting would be very interesting to find out.

2- The fact that adding more C3 data with noisy labels ends up damaging performance (l453) suggests an opportunity to treat the domain transfer as a semi-supervised problem (https://paperswithcode.com/sota/semi-supervised-image-classification-on-2) which would have the benefit of potentially enabling the use of much more unlabeled data. More generally, if a large pool of unlabeled data is available, training a self-supervised model using contrastive loss on it, and then fine-tuning it using labeled data is fast becoming the

Nit: l280 is missing a model name.

Reviewer #3: Rose N. et al reported a follow-up study to enhance the original MARCO model. The MARCO performs well with the labelled training set from the MARCO publication (Bruno AE, 2018), but it does not perform as accurately on local datasets, which restricts its use in the ‘real world’. The authors have conducted extensive experiments to identify training settings that could help MARCO perform better at a local environment using ML (machine learning). The proposed model with the most recent machine learning architectures improves the accuracy of crystallization drop analysis outside of the test datasets which were used for the 2018 MARCO paper. They also considered the previous limitations and performed a new evaluation of the datasets for the present study.

General questions/remarks:

(1) Using “local” set from the same lab (C3) as part of the test set to train MARCO may introduce bias, even when those sets have been labelled by different crystallographers.

(2) The performance of the new model is marginally better, but concerns about the quality and the diversity of the original MARCO images remain. Will it be more relevant to produce new and more diverse and controlled MARCO training and test sets?

(3) I am aware that sorting crystallization outcome is complex, but the “other” meta-label is too broad in my opinion. Poorly dispensed drops that are useless and drops with phase separation or spherulites that could be promising initial conditions may fall into this group. Can they be distinguished using the new ML model?

(4) Do the imager settings influence the success of MARCO training with local sets?

(5) Can the labeling differences between the test set and the local set affect the training's effectiveness? Since they were carried out by many individuals, the results' grading may vary.

(6) line 280: What is “XX” architecture?

6. PLOS authors have the option to publish the peer review history of their article (what does this mean?). If published, this will include your full peer review and any attached files.

Reviewer #1: No

Reviewer #2: No

Reviewer #3: No

---

## [Author Response · Author response to Decision Letter 0]

17 Feb 2023

Please see the attached "Response to reviewers" document that details the changes we have made to the manuscript in response to the editors and the reviews. We are grateful for the time and effort that has been put into reading and reviewing this work.

---

## [Editor Report · Decision Letter 1]

3 Mar 2023

Moving beyond MARCO

PONE-D-22-27063R1

Dear Dr. Newman,

We’re pleased to inform you that your manuscript has been judged scientifically suitable for publication and will be formally accepted for publication once it meets all outstanding technical requirements.

Kind regards,

Zhentian Wang, Ph.D.

Academic Editor

PLOS ONE

---

## [Editor Report · Acceptance letter]

16 Mar 2023

PONE-D-22-27063R1 

Moving beyond MARCO 

Dear Dr. Newman:

I'm pleased to inform you that your manuscript has been deemed suitable for publication in PLOS ONE. Congratulations! Your manuscript is now with our production department. 

Kind regards, 

on behalf of

Prof. Zhentian Wang 

Academic Editor

PLOS ONE